# Neuro-Argumentative Learning with Case-Based Reasoning

**Adam Gould**[*]                                        adam.gould19@imperial.ac.uk
and **Francesca Toni**                                        f.toni@imperial.ac.uk
*Department of Computing, Imperial College London, UK*

**Editors:** Leilani H. Gilpin, Eleonora Giunchiglia, Pascal Hitzler, and Emile van Krieken

## Abstract

We introduce *Gradual Abstract Argumentation for Case-Based Reasoning (Gradual AA-CBR)*, a data-driven, neurosymbolic classification model in which the outcome is determined by an argumentation debate structure that is learned simultaneously with neural-based feature extractors. Each argument in the debate is an observed case from the training data, favouring their labelling. Cases attack or support those with opposing or agreeing labellings, with the strength of each argument and relationship learned through gradient-based methods. This argumentation debate structure provides human-aligned reasoning, improving model interpretability compared to traditional neural networks (NNs). Unlike the existing purely symbolic variant, *Abstract Argumentation for Case-Based Reasoning (AA-CBR)*, Gradual AA-CBR is capable of multi-class classification, automatic learning of feature and data point importance, assigning uncertainty values to outcomes, using all available data points, and does not require binary features. We show that Gradual AA-CBR performs comparably to NNs whilst significantly outperforming existing AA-CBR formulations.

## 1. Introduction

Interpreting neural-based models is challenging because of their size, mathematical complexity and latent representations (Fan et al., 2020). Despite their success in classification tasks, it is unclear what lines of reasoning are applied to the data. On the other hand, symbolic AI represents knowledge bases as symbols that can be reasoned with in much the same way as humans. Computational argumentation, for example, resolves situations of uncertainty by applying human-aligned argumentative reasoning (Dung, 1995; Čyras et al., 2021). However, purely symbolic methods struggle to scale and generalise to large datasets, and cannot easily handle noisy or unstructured data. Neurosymbolic methods offer the best of both worlds (Garcez and Lamb, 2023).

To this end, we propose *Gradual Abstract Argumentation for Case-Based Reasoning (Gradual AA-CBR)*. This neurosymbolic model is the first-of-its-kind to learn the structure of a case-based debate via gradient-based methods (Rumelhart et al., 1986). Every data point in the training set argues in favour of its labelling, attacking those with an opposing label and supporting those with the same label. Gradual AA-CBR uses a neural-based feature extractor to learn the relative importance of each argument and how arguments should relate to each other. Figure 1 showcases an example of the architecture. Given the argument relationships, the final strength of each argument is computed and used to determine the outcome of a new, unlabelled data point (Baroni et al., 2018; Amgoud and Doder,

---

[*] Corresponding Author

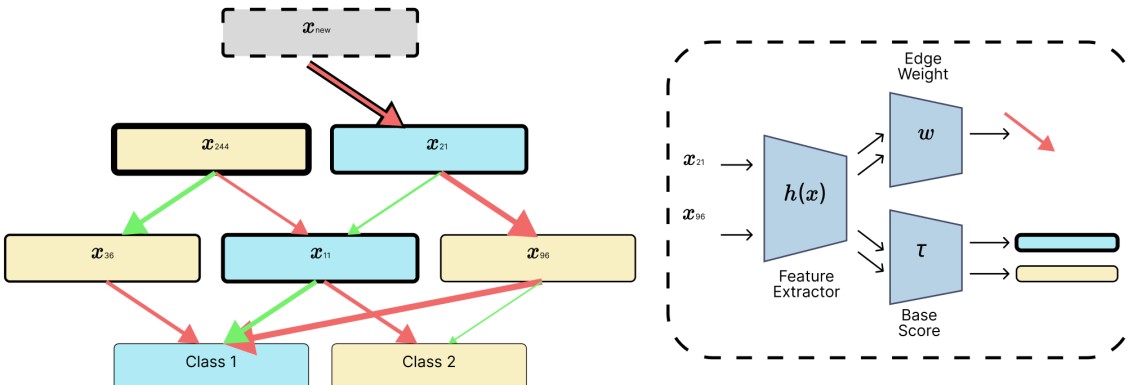

Figure 1: An example case-based debate generated by Gradual AA-CBR. Each node in the graph on the left-hand side is an argument represented by a data point in the training set. Red and green arrows indicate attacks and supports, respectively. The thickness of the argument borders and arrows represents the strength of the arguments and relationships. Argument importance and relationship strengths are computed with the feature extractor on the right-hand side.

2019). We can apply a gradient descent-like algorithm to optimise the model parameters, provided these final strengths are computed with differentiable functions. Unlike neural network (NN) inference, this reasoning process is transparent and interpretable.

Furthermore, this enables Gradual AA-CBR to overcome the limitations of the purely symbolic variant *Abstract Argumentation for Case-Based Reasoning (AA-CBR)* (Čyras et al., 2016), allowing it to i) do multi-class classification, improving the applicability of the model; ii) automatically learn which features and data points are essential, leading to improved classification performance; iii) assign a quantified acceptability score to each argument, allowing users to calibrate their trust in the prediction; and iv) operate beyond binary features, expanding the types of data that the model can be applied to. Our contributions are summarised as follows[1]:

1. We introduce Gradual AA-CBR, which quantifies a case-based debate, allowing for argumentative reasoning with degrees of acceptability.

2. We show how Gradual AA-CBR can be parametrised with learnable weights that can be updated via gradient-based methods. This allows for end-to-end learning of the Gradual AA-CBR structure using neural-based feature extractors.

3. We show that Gradual AA-CBR achieves performance comparable to neural-based models without sacrificing interpretability yet outperforms other AA-CBR variants.

In Section 2 we discuss related works. Then, in Section 3 we provide the necessary background before introducing our neurosymbolic model in Section 4. We demonstrate the effectiveness of our approach in Section 5 and conclude in Section 6.

---

1. Experiment code is available at: https://github.com/TheAdamG/Gradual-AACBR

## 2. Related Work

AA-CBR (Čyras et al., 2016) is a purely symbolic model that uses argumentation with previously observed data points to make a binary classification for a new, unlabelled data point. AA-CBR treats data points as observed *cases*, which state a general rule in favour of their labelled outcome. Each case may have multiple *exceptions*, which are represented by an *attack* relationship between the cases. The most general rule is the *default case*, which is associated with a *default outcome* that the model predicts when unable to make a prediction with the observed cases. To classify a new data point, cases *irrelevant* to it are excluded, and the argumentation debate is resolved to determine if the default outcome is accepted. Unlike Gradual AA-CBR, this model only does binary classification, cannot determine feature or data point importance, will not quantify its confidence in its output and struggles to handle data types more complex than binary features.

Preference-Based AA-CBR (AA-CBR-$\mathcal{P}$) (Gould et al., 2024) is a variant of AA-CBR that introduces user-defined preferences over features. This method improved performance in a medical application with clinical knowledge injected and thus shows the promise of AA-CBR with feature importance. However, this approach requires manually specifying these preferences. Deciding what features are preferable or optimised for the classification task may not be obvious. Gradual AA-CBR automatically optimises feature importance.

Neuro-argumentative Learning (NAL) studies how to combine argumentation with neural-based models (Proietti and Toni, 2023). Applications include argumentation-to-NN translation (d'Avila Garcez et al., 2005, 2014), constraining the learning of neural-based models (Riveret et al., 2020) and argument mining (Cocarascu et al., 2019; Freedman et al., 2024). Gradual AA-CBR differs from these, learning end-to-end with gradient-based techniques using a quantitative argumentation framework as the underlying structure.

One such NAL method, *Artificial Neural Networks for Argumentation (ANNA)*, uses AA-CBR as a classifier of features extracted from an NN (Cocarascu et al., 2020, 2018). This approach operates in a pipeline fashion, wherein the NN (e.g. an autoencoder (Bank et al., 2023)) is trained separately from the AA-CBR model. However, this approach cannot learn features that will be the best for argumentative reasoning. The AA-CBR component of the pipeline is still purely symbolic, so all limitations outlined above apply.

Another NAL approach interprets feed-forward NNs as argumentation frameworks to understand or simplify their structure (Potyka, 2021; Ayoobi et al., 2023b). This has been applied to prototype networks for image classification, where prototypical images can be recalled as the reason for the assigned class (Ayoobi et al., 2023a) in a different formulation of case-based reasoning to AA-CBR. However, this approach still contains hidden neurons as part of the final component that classifies the similarities to the prototypes, hindering the model's overall interpretability. Our approach uses data points rather than hidden neurons in the argumentation debate and thus does not have this limitation.

As Gradual AA-CBR learns the structure of the debate, we can look to Graph Structure Learning (GSL), a subfield of Graph Neural Networks (GNN) that optimises the structure of a graph simultaneously with the network weights for a downstream classification task (Zhu et al., 2021). Similar to Jiang et al. (2019); Luo et al. (2021), we allow an NN to discover the relationships between nodes, although we do not use a GNN as the underlying structure. As in Luo et al. (2021), we use a regulariser that encourages community preservation.

## 3. Background

Gradual AA-CBR is built upon the Edge-Weighted Quantitative Bipolar Argumentation Framework (Potyka, 2021).

**Definition 1 (Edge-Weighted Quantitative Bipolar Argumentation Framework)**
*An edge-weighted QBAF is a quadruple $\langle \mathcal{A}, \mathcal{E}, \tau, w \rangle$, where $\mathcal{A}$ is a set of arguments, $\mathcal{E}$ is a set of edges between arguments, $\tau : \mathcal{A} \to [0, 1]$ is a total function that maps every argument $a \in \mathcal{A}$ to a* base score *and $w : \mathcal{E} \to \mathbb{R}$ is a total function that maps every edge to a weight[2].*

Intuitively, the base score of an argument can be interpreted as its intrinsic strength before we consider its relationships to other arguments. For instance, a base score may be the uncertainty that the argument represents a truthful statement or our initial belief of the degree of acceptability. Similarly, edge weights define the strength of the arguments' relationships, with negative weights representing an *attack* and positive weights a *support*. We can represent these graphically as in Figure 1.[3]

We must consider how the strengths of the arguments change when they have been attacked or supported. For example, if an argument $a$ with a base score of 0.5 is attacked with a strength of -1 by an argument $b$ with a base score of 1, then we expect the final strength of $a$ to decrease, say to 0.25, signifying that the acceptability of $a$ has reduced. The approach used to compute the degree of acceptability is called the *gradual semantics* (Amgoud and Doder, 2019; Baroni et al., 2018). The *final strength* of an argument $a \in \mathcal{A}$ in an edge-weighted QBAF is given by the function $\sigma : \mathcal{A} \to [0, 1] \cup \{\perp\}$, where $\perp$ means the final strength is undefined.

*Modular semantics* (Mossakowski and Neuhaus, 2018; Potyka, 2019), involves iteratively computing the acceptability of each argument through *aggregation* of the strengths of attacks and supports whilst accounting for the *influence* of the arguments' base scores. For this work, we focus on semantics that simulate a multi-layer perceptron (MLP) (Potyka, 2021) as this considers the edge weights and is differentiable, which will be necessary for gradient-based optimisation techniques. MLP semantics are defined as follows:

**Definition 2** *Let $\psi_a^{(i)} \in [0, 1]$ be the strength of argument $a$ at iteration $i$. For every argument $a \in \mathcal{A}$, we let $\psi_a^{(0)} := \tau(a)$ so that the starting strength is equal to the base score. The strength values are computed by repeating the following two functions:*

**Aggregation:** *Let $\rho_a^{(i+1)} := \sum_{(b,a)\in\mathcal{E}} w(b,a) \cdot \psi_b^{(i)}$*

**Influence:** *Let $\psi_a^{(i+1)} := \varphi(\varphi^{-1}(\tau(a)) + \rho_a^{(i+1)})$, where $\varphi$ is a non-linear activation function.*

*The final strength of argument $a$ is given by $\sigma(a) = \lim_{k\to\infty} \psi_a^{(k)}$ if the limit exists and $\perp$ otherwise.*

In this work, as in work by Ayoobi et al. (2023b), we let the activation be the ReLU, that is $\varphi = max(0, x)$. Though not invertible, we let $\varphi^{-1} = max(0, x)$.

---

2. With a slight abuse of notation we remove the inner brackets, such that $w((a,b))$ becomes $w(a,b)$.

3. An edge weight of 0 between a given pair of arguments is semantically equivalent to no edge between these nodes. We need only consider those with non-zero edge weights.

## 4. Methodology

### 4.1. Gradual AA-CBR

We now introduce Gradual AA-CBR. Intuitively, Gradual AA-CBR reasons analogously to AA-CBR (see Section 3), with previously observed data points forming a casebase. However, in Gradual AA-CBR cases with the same outcome may also *support* other cases. The functions $\tau_x$ and $w_{\succcurlyeq}$ are provided to determine the argument base scores and edge weights. Instead of a single default argument, which could bias the model towards an outcome, we introduce a set of *target cases*, each associated with a *target outcome*. Irrelevance to the new case is now defined by the function $w_{\nsim}$, and the argumentation debate is resolved with gradual semantics to determine the final strength of the target arguments. The model classifies the new case based on the outcome of the target argument with the greatest final strength. Now we can define Gradual AA-CBR as follows:

**Definition 3 (Gradual AA-CBR)** *Let $D \subseteq X \times Y$ be a finite* casebase *of labelled examples where $X$ is a set of* characterisations *and $Y = \{c_1, \ldots, c_m\}$ ($m \geq 2$) is the set of possible outcomes. Each data point is of the form $(x_a, y_a)$. Let $\mathcal{T} = \{(x_{\delta_i}, c_i) \mid c_i \in Y\}$ be the set of* target arguments *corresponding to each class label, with $x_{\delta_i}$ the default characterisation for class $c_i$. Let $\mathcal{A}' = D \cup \mathcal{T}$ be the set of labelled cases. Let $N$ be an unlabelled example of the form $(x_N, y_?)$ with $y_?$ an unknown outcome. Let $\tau_x : X \to [0,1]$ be a function mapping characterisations to base scores. Let $w_{\succcurlyeq} : X \times X \to [0,1]$ and $w_{\nsim} : X \times X \to [-1,0]$ be functions mapping pairs of characterisations to weights. The edge-weighted QBAF mined from $D$ and $x_N$ is $\langle \mathcal{A}, \mathcal{E}, \tau, w \rangle$ in which:*

- $\mathcal{A} = \mathcal{A}' \cup \{N\}$,

- $\tau((x_a, y_a)) = \tau_x(x_a)$,

- $\mathcal{E} = \{((x_a, y_a), (x_b, y_b)) \in \mathcal{A}' \mid a \neq b\} \cup$
  $\quad \{((x_N, y_?), (x_a, y_a)) \mid (x_a, y_a) \in \mathcal{A}'\}$,

- $w((x_a, y_a), (x_b, y_b)) = \begin{cases} w_{\nsim}(x_N, x_b) & \text{if } (x_a, y_a) = (x_N, y_?), \\ w_{\rightsquigarrow}((x_a, y_a), (x_b, y_b)) & \text{if } y_a \neq y_b, \\ w_{\rightarrow}((x_a, y_a), (x_b, y_b)) & \text{otherwise}, \end{cases}$

  *where*

  - $w_{\rightsquigarrow}((x_a, y_a), (x_b, y_b)) = -[w_c(x_a, x_b, \mathcal{F}(\mathcal{A}', y_a)) + w_=(x_a, x_b)]$,
  - $\mathcal{F}(S, y_a) = \{(x_c, y_c) \in S \mid y_a = y_c\}$
  - $w_{\rightarrow}((x_a, y_a), (x_b, y_b)) = w_c(x_a, x_b, \mathcal{A}')$,
  - $w_c(x_a, x_b, S) = w_{\succ}(x_a, x_b) \cdot \prod_{(x_c, y_c) \in S} (1 - (w_{\succ}(x_a, x_c) \cdot w_{\succ}(x_c, x_b)))$.
  - $w_=(x_a, x_b) = w_{\succcurlyeq}(x_a, x_b) \cdot w_{\succcurlyeq}(x_b, x_a)$,
  - $w_{\succ}(x_a, x_b) = w_{\succcurlyeq}(x_a, x_b) \cdot (1 - w_{\succcurlyeq}(x_b, x_a))$,

Intuitively, we define our set of arguments as the observed cases, the target arguments, and the new, unlabelled case. The base score of each case is determined by a provided function $\tau_x$, depending only on the argument's characterisation. We define edges between every pair of distinct labelled arguments or from the new case to every labelled argument.

The weight of the edges is determined by one of three ways. Firstly, a provided $w_\nsim$ function is used to determine the strength of the irrelevance attacks by the new case. Secondly, for arguments with different outcomes, the strength of attacks is given by $w_\rightsquigarrow$, which is computed using a provided $w_\succcurlyeq$ function. $w_\succcurlyeq$ defines the degree of exceptionalism between cases in the casebase, with $w_\succcurlyeq(x_a, x_b) = 1$ meaning argument $a$ is considerably more exceptional than $b$, whereas $w_\succcurlyeq(a, b) = 0$ means $a$ is not more exceptional than $b$. $w_\succ$ and $w_=$ define strict exceptionalism and equal exceptionalism, respectively, such that cases with the same level of exceptionalism attack each other symmetrically; otherwise, strictly more exceptional cases attack less exceptional cases with greater strength than in reverse. The function $w_c$ enforces a soft *minimality* constraint, such that an attacking case $a$ is most minimal to an attacked case $b$ if there is not another case, $c$ with the same outcome as $a$ that is more similar to $b$, thus ensuring cases most similar to those they attack have larger magnitude weights.

Finally, the edge weights for supports, given by $w_\rightarrow$, is defined similarly for cases with the same outcome with two minor changes: i) as we have sets of data points, there cannot be two cases with the same characterisation and outcome, so we need not apply $w_=$, and ii) a supporting case $a$ is most minimal to a supported case $b$ if there is not another case, $c$ with *any* outcome that is more similar to $b$. We do not enforce that $c$ must have the same outcome as $a$ so that similarity between cases with opposing outcomes will also cause the weight of the support to decrease, allowing attacks to have priority over supports.

Furthermore, unlike with a traditional edge-weighted QBAF we enforce that the range of $w$ is $[-1, 1]$. To do so, we should constrain $w_\succcurlyeq$ such that $w_\succ$ and $w_=$ have the range $[0, 1]$. We therefore enforce that:

$$w_\succcurlyeq(x_a, x_b) = 1 - w_\succcurlyeq(x_b, x_a). \tag{1}$$

Additionally, we enforce some constraints on the choice of $\mathcal{T}$ and $w_\nsim$ to guarantee behaviour that is consistent with reasoning intuitions we expect:

**Definition 4 (Regular Gradual AA-CBR)** *The edge-weighted QBAF mined from D and N, with target arguments* $\mathcal{T} = \{(x_{\delta_c}, y_c) \mid y_c \in Y\}$ *is* regular *when:*

1. *$w_\nsim(N, (x_a, y_a)) = w_\succcurlyeq(N, (x_a, y_a))$ - 1*

2. *$\forall k, l \in [1, m], x_{\delta_k} = x_{\delta_l}$ and $\forall x_a \in X, w_\succcurlyeq(x_a, x_{\delta_k}) = 1$ and $w_\succcurlyeq(x_{\delta_k}, x_a) = 0$.*

Condition (1) ensures that $w_\nsim$ is defined in terms of $w_\succcurlyeq$. This is crucial when we allow $w$ to contain parameters learned by gradient-based methods as described in Section 4.2. If $w_\nsim$ was unrelated to $w_\succcurlyeq$, it may learn that new cases should only attack target arguments and ignore the arguments in the casebase and thus will not apply case-based reasoning.

Condition (2) enforces that all target arguments have the same characterisation and that every case in the casebase will always be considered more exceptional than the target arguments. This ensures that the target arguments cannot attack/support other arguments and that the QBAF always directs chains of attacks/supports towards the target arguments. From now on, we will be using Regular Gradual AA-CBR unless specified otherwise.

## 4.2. An End-to-End Neuro-Argumentative Learner

For a new case, $N$, we predict the outcome $y_?$ by applying a gradual semantics on the constructed QBAF and selecting the target argument with the maximum final strength. Formally:

$$\text{gAA-CBR}(D, N) = \underset{c_i \in Y}{\text{argmax}}\, \sigma((x_{\delta_i}, c_i)), \tag{2}$$

where $\sigma$ is a gradual semantics such as that in Definition 2.

### 4.2.1. FUNCTION CHOICES

Choosing a gradual semantics that is differentiable, for example, MLP-based semantics (Definition 2) and selecting differentiable functions for $\tau_x$ and $w_{\succcurlyeq}$ means these can contain parameters learnable by gradient-based methods. These functions could be set to any neural-based model such as an NN, convolutional neural network (CNN) (LeCun et al., 1989, 1998), or transformer (Vaswani et al., 2017). Furthermore, the parameters used by $\tau_x$ and $w_{\succcurlyeq}$ can be shared, ensuring that the same features extracted are used for both.
This work focuses on real-valued data where each $x_a$ is of the form $[x_{a,1}, x_{a,2}, ..., x_{a,d}]^\top \in \mathbb{R}^d$. For feature extraction, we use the function,

$$h(x_a) = \sum_{k=1}^{d} \theta_k x_{a,k} \tag{3}$$

where $\theta_k$ is a learnable weight for feature $x_{a,k}$. We then define the base score and edge weight functions as

$$\tau_x(x_a) = S(h(x_a) \cdot \theta_s) \tag{4}$$

$$w_{\succcurlyeq}(x_a, x_b) = S((h(x_a) - h(x_b)) \cdot T) \tag{5}$$

where $S(x) = \frac{1}{1+exp(-x)}$ is the sigmoid function, $\theta_s$ is a learned scale value and $T$ is a temperature hyperparameter.

Equation (3) defines a function that weights each feature of $x_a$. These weights are shared for $\tau_x$ and $w_{\succcurlyeq}$ such that relative feature contributions are the same for both base scores and the edge weights, even if the absolute contributions can be scaled by $\theta_s$. Equation (5) defines an approximate partial order such that if $h(x_a) > h(x_b)$, then $w_{\succcurlyeq}$ will return a value closer to 1 and a value closer to 0 otherwise. The sigmoid function, $S$, ensures that the ranges of $\tau_x$ and $w_{\succcurlyeq}$ are $[0, 1]$ and that the constraint defined in Equation (1) holds.

When $w_{\succcurlyeq}(x_a, x_b)$ returns a value between 0 and 1 (exclusive), then by Equation (1), there will always be a non-zero value in the reverse direction. This leads to a situation where for every pair of arguments, $a, b \in \mathcal{A}'$, there is an edge from $a$ to $b$ and from $b$ to $a$ both with non-zero edge weights in the resulting QBAF. Cycles in the QBAF are problematic as the model becomes difficult to interpret, and, as identified by Potyka (2021), may prevent the semantics from converging. To remedy this, we apply a post-process function, $\mathcal{P}$, defined using the indicator function $\mathcal{X}(B) = 1$ if $B$, and 0 otherwise, formally:

$$\mathcal{P}(a, b) = w(a, b) \cdot \mathcal{X}(|w(a,b)| \geq |w(b,a)|) \tag{6}$$

which for cyclic edges between pairs of arguments, ensures that only the edge with a larger weight is kept and the other is set to 0.

### 4.2.2. MODEL TRAINING

The model can be trained for a classification task by minimising the categorical cross-entropy loss function $\mathcal{L}_{ce}$ (Ciampiconi et al., 2023). Furthermore, we can apply regularisation during training to enforce soft constraints on the graphical structure of the learned QBAF. To do so, we can represent the post-processed edge weight function $\mathcal{P}$ as an adjacency matrix $\mathbf{A}$ wherein $\mathbf{A}_{i,j} = \mathcal{P}(i,j)$ where $i$ and $j$ correspond to the $i$-th and $j$-th argument of $\mathcal{A}'$ respectively. Then, as in existing GSL approaches (Zhu et al., 2021), we apply the community preservation regularisation $\mathcal{L}_{cp} = \text{rank}(\mathbf{A})$. It acts as a soft constraint, causing the graph to have better-defined communities, which is beneficial for ensuring the most critical arguments are highly connected. The model is, therefore trained by minimising the following function:

$$\mathcal{L} = \mathcal{L}_{ce} + \gamma\mathcal{L}_{cp} \tag{7}$$

where $\gamma \geq 0$ is a hyperparameter controlling the trade-off between classification and regularisation loss[4].

The training algorithm is based on gradient descent and can be found in the supplementary material. The primary difference when training Gradual AA-CBR compared to gradient descent is the need for a fit step in which we first construct the QBAF with the casebase. Then, as in traditional NN training, we iterate through the training data, making a class prediction for each data point and backpropagating the loss gradients. However, in Gradual AA-CBR, the subject data point must be treated as a new case with a new QBAF constructed, and the forward pass involves computing the gradual semantics. Besides these changes, we update parameters based on the loss gradient as in gradient descent.

A suitable choice for the default characterisation that ensures we obey Regular Gradual AA-CBR (Definition 4) is to set $x_{\delta_c}$ to the mean characterisation of the training data, that is, we let $x_{\delta_c} = \mu(X_t)$ where $X_t = \{x_a \mid (x_a, y_a) \in D\}$. When the dataset is normalised, $\mu(X_t)$ is equal to the zero vector and so $x_{\delta_c} = \mathbf{0}$. With our choice of base score function (Equation (4)), we therefore have $\tau_x(x_{\delta_c}) = 0.5$, which ensures the base score is not initially biased towards any one outcome.

## 5. Experiments

We evaluate Gradual AA-CBR using the standard classification metrics, accuracy and macro averaged precision, recall and f1 score (Hossin and Sulaiman, 2015). We experiment with three binary classification datasets, Mushroom (Unknown, 1981), Breast Cancer (Wolberg et al., 1993), and Glioma Grading (Erdal Tasci, 2022), and one multi-class classification set, Iris (R. A. Fisher, 1936). Each experiment is evaluated against a simple NN with no hidden layers, and for the Mushroom and Glioma datasets, which contain only binary features, we test against AA-CBR and ANNA[5].

---

4. In practice, we can regularise the attacks (negative edge weights) and supports (positive edge weights) independently, leading to more stable training.

5. A discussion of dataset details and selected model hyperparameters for each dataset and baseline is provided in the supplementary material.

Table 1: Classification results across four datasets

| Model | Accuracy | Precision | Recall | $F_1$ Score |
|---|---|---|---|---|
| **Mushroom** | | | | |
| AA-CBR | 0.48 | 0.24 | 0.50 | 0.33 |
| ANNA | 0.84 | 0.85 | 0.84 | 0.84 |
| NN | 0.99 | 0.99 | 0.99 | 0.99 |
| Gradual AA-CBR | 0.98 | 0.98 | 0.98 | 0.98 |
| **Glioma** | | | | |
| AA-CBR | 0.67 | 0.67 | 0.67 | 0.67 |
| ANNA | 0.67 | 0.67 | 0.67 | 0.67 |
| NN | 0.85 | 0.86 | 0.86 | 0.85 |
| Gradual AA-CBR | 0.86 | 0.86 | 0.86 | 0.86 |
| **Breast Cancer** | | | | |
| NN | 0.98 | 0.98 | 0.98 | 0.98 |
| Gradual AA-CBR | 0.95 | 0.95 | 0.94 | 0.94 |
| **Iris** | | | | |
| NN | 0.97 | 0.97 | 0.96 | 0.97 |
| Gradual AA-CBR | 0.97 | 0.97 | 0.96 | 0.97 |

### 5.1. Experimental Results and Discussion

Table 1 shows the classification performance observed on the test set across the four datasets. Gradual AA-CBR performs comparably to the NN on all datasets, with only small margins of performance lost. On the datasets with binary features, Gradual AA-CBR considerably outperforms both AA-CBR and ANNA. We observe deficiencies in the previous models that Gradual AA-CBR can overcome. Firstly, ANNA is not a catch-all for AA-CBR as we see with the Glioma dataset in which no subset of features was found that could perform better than using all features. Secondly, we did not run AA-CBR and ANNA on the Breast Cancer and Iris datasets as they contain continuous features and defining a notion of exceptionality over these becomes exceedingly difficult. Finally, AA-CBR and ANNA only work for binary classification, and so even if we could find such a notion, these models would still not work for Iris. Table 2 summarises these model capabilities, wherein only Gradual AA-CBR achieves high performance whilst being interpretable, capable of multi-class classification and using continuous data.

The interpretability of Gradual AA-CBR is the key highlight of the model. We can visualise the QBAF and inspect the features weights, edge weights and case base scores[6]. This is a significant advantage over an NN, wherein even small models are challenging to interpret.

---

6. The supplementary material includes a full figure of a learned QBAF for the Iris dataset.

Table 2: Model Capabilities

| Model | Performant | Interpretable | Multi-class | Continuous Data |
|---|---|---|---|---|
| AA-CBR | × | ✓ | × | × |
| ANNA | ∼ | ✓ | × | × |
| NN | ✓ | × | ✓ | ✓ |
| Gradual AA-CBR | ✓ | ✓ | ✓ | ✓ |

× - model is not capable, ∼ - model is partially capable, ✓ - model is capable

Whilst we present the best-observed results here, a notable weakness of Gradual AA-CBR is that the model performance is sensitive to the choice of initial weights. For a simple NN used for classification, weights are initially randomised, typically with an initialisation function imposing some constraints on the values that the weights can take. This leads to the NN randomly predicting each class an equal number of times before training. For Gradual AA-CBR, the model typically starts by predicting every instance as a single class. Thus, training can quickly get stuck in a local minimum where the logits for a single class are correctly minimised, but logits for other classes are not. As the number of possible initial states scales with the number of features of the dataset, this limitation can prevent the model from scaling to high-dimensional data, and future work must look at new initialisation schemes. AA-CBR on the Iris dataset can learn in 79% of initial states tried. The key features that helped improve this rate is using Xavier Normal initialisation (Glorot and Bengio, 2010), introducing supports, using graph-based regularisation and hyperparameter tuning of the learning rate, number of epochs and optimiser. However, we found that on the Breast Cancer and Glioma datasets, the model can only learn in 16% and 11% of initial states, prompting future work on the matter.

## 6. Conclusion

We have introduced Gradual AA-CBR, a novel neuro-argumentative learning model capable of leveraging the reasoning capabilities of AA-CBR with feature extractors learned end-to-end. This model successfully matches the performance of an NN on continuous data whilst providing interpretable and transparent reasoning. There are many avenues for future work, including a richer analysis of the intepretability of the model, developing a new initialisation scheme to improve the percentage of initial states the model can learn from, experimenting with feature extraction on more complex tasks and data types, for instance images and time series using more complex feature extractors such as CNNs (LeCun et al., 1989, 1998), RNNs (Lipton et al., 2015) or transformers (Vaswani et al., 2017), and applying explainable AI techniques to extract tailored explanations of the model (Čyras et al., 2021). Furthermore, other variants of AA-CBR, such as cumulative AA-CBR (Paulino-Passos and Toni, 2021) and preference-based AA-CBR (Gould et al., 2024) could be adapted to neuro-argumentative learning models in much the same way as our method, thus allowing for monotonic reasoning methods or user-injected preferences.

## Acknowledgments

Research by Adam Gould was supported by UK Research and Innovation [UKRI Centre for Doctoral Training in AI for Healthcare grant number EP/S023283/1]. Francesca Toni was partially funded by the ERC under the EU's Horizon 2020 research and innovation programme (grant agreement No. 101020934). Toni was also partially funded by J.P. Morgan and the Royal Academy of Engineering, UK, under the Research Chairs and Senior Research Fellowships scheme.

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

## Appendix A. Gradual AA-CBR Training

As described in the main body, Algorithm 1 showcases the training algorithm for Gradual AA-CBR. The primary difference when training Gradual AA-CBR compared to a traditional NN is the need for a fit step (line 4) in which we first construct the QBAF with the casebase, referred to as $QBAF_d$. Then, as in traditional NN training, we iterate through the training data, making a class prediction for each data point and propagating the loss. However, in Gradual AA-CBR, the subject data point must be treated as a new case with a new QBAF constructed (line 6) and the forward pass involves computing the gradual semantics (line 7). Otherwise, we propagate the loss and update parameters as standard.

---

**Algorithm 1** Gradual AA-CBR Training with Gradient Descent

---

1. **Input:** Training data $D_t$, learning rate $\alpha$, number of epochs $E$, semantics $\sigma$, base score function $\tau_x$, edge weight functions $w_{\succcurlyeq}$ and $w_{\nsim}$, target arguments $\mathcal{T}$,

2. **Initialize:** Function parameters $\theta$ of $\tau_x$, $w_{\succcurlyeq}$ and $w_{\nsim}$ randomly or using a specific initialization method

3. For epoch $= 1$ to $E$

    4. Fit the QBAF on the training data $D_t$, such that $F := QBAF_{D_t}$

    5. For each case $a := (x_a, y_a)$ in $D_t$

        **Forward Pass:**

    6. Add case $a$ as a new case to $F$, giving $F' := QBAF_{D_t, a}$

    7. Compute the output $\hat{y}_a := [\sigma(t_1), \sigma(t_2), ..., \sigma(t_m)]^\top$, for each target argument $t_i$ of $F'$

    8. Compute loss $\mathcal{L}(\hat{y}_a, y_a)$

        **Backward Pass:**

    9. Compute the gradient of the loss with respect to the parameters $\nabla_\theta \mathcal{L}(\hat{y}_a, y_a)$

        **Update Parameters:**

    10. Update parameters: $\theta := \theta - \alpha \nabla_\theta \mathcal{L}(\hat{y}_a, y_a)$

11. **Output:** Trained weights $\theta$

---

## Appendix B. Experiment Details

The baseline NN for all models was a single layer NN with an input size equal to the number of features in the dataset and the output size is the number of classes. It was trained with Categorical Cross Entropy Loss (Ciampiconi et al., 2023), using the Adam optimiser Kingma and Ba (2014) with a learning rate of 0.02 and 500 epochs.

The Mushroom dataset is used for binary classification, distinguishing poisonous or edible mushrooms. It consists of 8124 instances and 117 one-hot encoded binary features and was used to compare Gradual AA-CBR against ANNA and AA-CBR with exceptionalism defined by the subset relation as in Cocarascu et al. (2018). For ANNA, an autoencoder with a hidden layer size of 30 was used to select the top 22 features. We use 200 randomly chosen instances for model training, 50 for validation, and 7874 as the test set. Gradual AA-CBR was trained with the Adam Optimiser, with a learning rate of 0.02 for 500 epochs. The temperature $T$ was set to 0.05, and $\gamma$ was set to 0.005.

Similarly, the Glioma dataset is used for binary classification, grading brain tumours as either low or high grade. It consists of 839 instances with 21 binary features[7]. 364 randomly chosen instances were used for model training, 91 for validation and 114 for the test set.

---

7. There is also one real-valued feature, patient age, and one categorical feature, race, which we exclude as AA-CBR/ANNA require binary features.

For ANNA, an autoencoder with multiple hidden layer sizes, 5, 10, 15, 30 was tried, but no subset of features found lead to better results than using all features as with AA-CBR. Gradual AA-CBR was trained with the Adam Optimiser, with a learning rate of 0.02 for 6000 epochs. The temperature $T$ was set to 0.05, and $\gamma$ was set to 0.005.

The Breast Cancer dataset is used for binary classification, distinguishing malignant vs benign tumours. It consists of 569 instances with 30 real-valued features. 364 randomly selected instances were used for model training, 91 for validation and 114 for the test set. Gradual AA-CBR was trained with the Adam Optimiser, with a learning rate of 0.02 for 2500 epochs. The temperature $T$ was set to 0.05, and $\gamma$ was set to 0.005.

Finally, the Iris dataset was selected to demonstrate the ability to expand to multi-class classification. This dataset consists of 150 instances of three classes with 4 continuous-valued features. 96 instances were randomly selected for training, 24 for validation and 30 for the test set. Gradual AA-CBR was trained with the Adam Optimiser, with a learning rate of 0.02 for 3000 epochs. The temperature $T$ was set to 0.05, and $\gamma$ was set to 0.005.

## B.1. Visualising the Learned QBAF

Figure 2 showcases a learned QBAF for the Iris dataset. We can see, for example, that the 28th data point is highly attacking; thus, the model considers this data point highly exceptional. As a result, the default for Class 2 is not strongly attacked or supported, suggesting that the model may, by default, predict a new data point as Class 2 unless node 28 is considered irrelevant to the new data point.

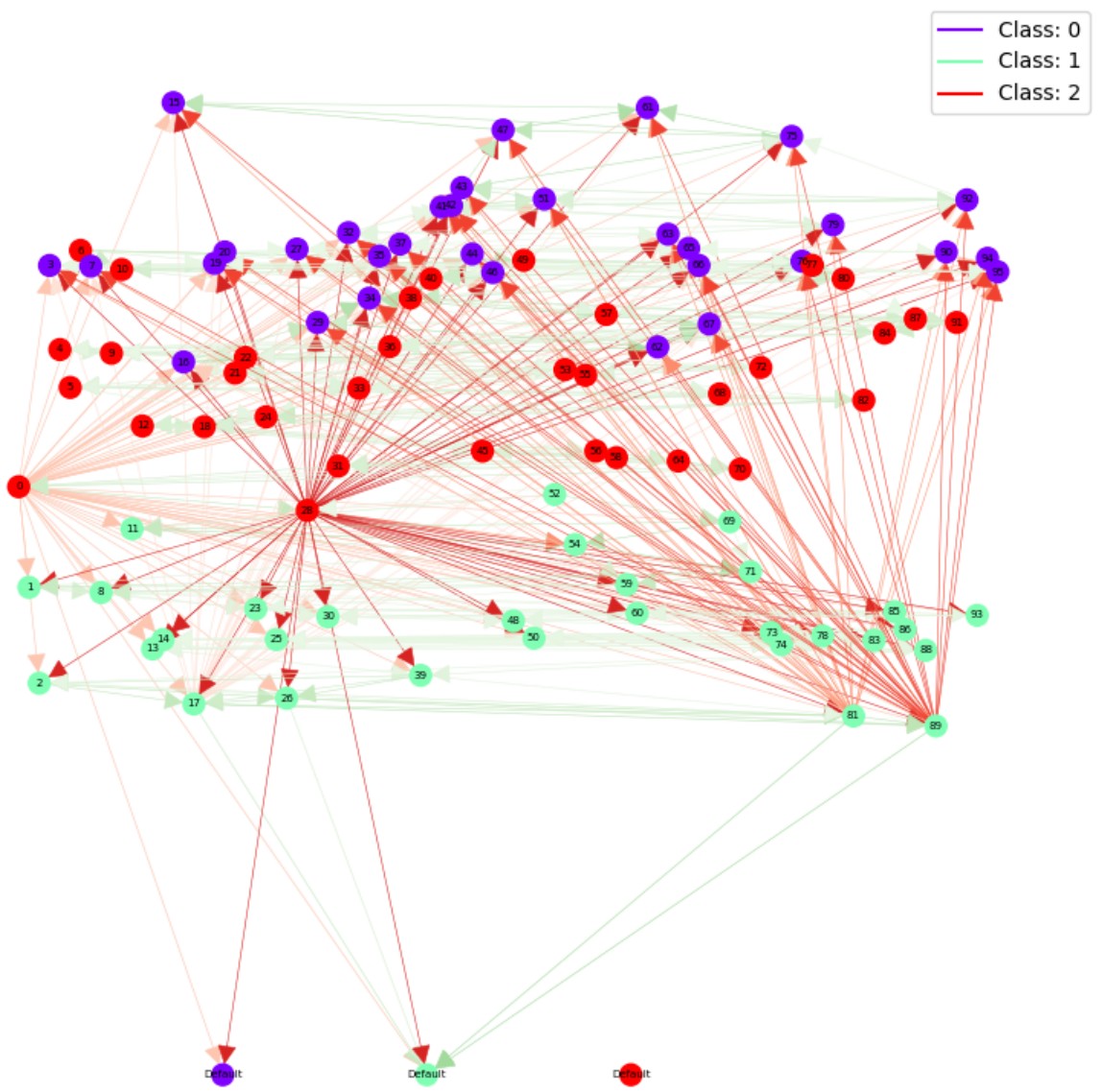

Figure 2: A learned QBAF for the Iris dataset. Every argument in the casebase is a node in the graph, with the edges from each node representing attacks (in red) or supports (in green). We filter the edges to only those with a magnitude greater than 0.1 for visualisation purposes. The intensity of the colour indicates the strength of the attack or support.

