# OpenReview forum: "Neuro-Argumentative Learning with Case-Based Reasoning"
_nesyconf.org/NeSy/2025/Conference — NeSy 2025 Poster_

### Official Review · Reviewer_QhZ2 · 2025-04-04
**Neuro-Argumentative Learning with Case-Based Reasoning - review**

**Rating:** 9
**Confidence:** 5

**Review:**

The authors describe an approach called Gradual Abstract Argumentation for Case-Based Reasoning (AA-CBR). Their main contributions are: an abstract argumentation CBR methodology that estimates quantitatively arguments and acceptability;  they show how gradual AA-CBR can deploly weights that are updates by gradient descent in end2end learning, and finally how their approach performs comparably with neural network based methods.
The paper builts upon ideas on neural-symbolic frameworks, in particular argumentation neural newtworks and CBR approaches. Thus, the paper is central to to the conference theme. The experiemtal part is also illustrative as they evaluate gradual AACBR using typical classification metrics, including recall and F1 scores and evaluate their setting against a simple connectionist network with no hidden layers. In conclusion of the experimental and foundational work, they show that their gradual AACBR approach can leverage reasoning of AA-CBR with feature extractors learned end2end.  The advantage of their aporoach resides in a sound and trasparent reasoning approach to argumentation.
Overall, this is a vey interesting paper that builds upon the foundations of argumentation neural networks, bringing case based-reasoning frameworks to the fore. I enjoyed reading the paper.

**Anonymity:**

Remain anonymous

---

### Official Review · Reviewer_amyq · 2025-04-05
**The neurosymbolic model Gradual Abstract Argumentation for Case-Based Reasoning (Gradual AA-CBR), that mixes argumentation and neural networks, is presented in this paper. Its main advantage is the ability to learn an argumentation debate structure, which improves interpretability. Although the methodology is generally straightforward, the formal definitions are complicated, and the interpretability claims need more careful evaluation.   Though the work is important for the development of explainable AI, challenges with scalability and the model's vulnerability to setup need more research because they could prevent its wide application. Therefore I rate this paper with a 7.**

**Rating:** 7
**Confidence:** 3

**Review:**

## Methodology

### Combines Argumentation and Neural Networks

It's a methodologically sound and novel decision to combine abstract reasoning with neural-based feature extractors. The authors clearly claim that this combination enables the machine to learn relationship and argument weights from data. However, there are two sides to this integration's complexity. It offers flexibility, but it also adds more variables and possible areas of failure.

### Employs Gradual Semantics

Gradual semantics is a useful addition that offers a more comprehensive method of evaluating arguments. Though justified for distinction, the MLP semantics may not be the most logical or natural choice for arguments.

### Learns Feature Importance
The authors clearly note that one of the model's major advantages is its capacity to automatically learn a set of features and data items.   The reliability and consistency of this , however, have not seem to  be thoroughly investigated.   Do various runs of the model consistently identify the same key features? To what extent does this feature selection keep up with faulty or noisy data?

### Addresses Limitations of Symbolic AA-CBR

The research paper makes it apparent that Gradual AA-CBR goes around a number of disadvantages of merely symbolic AA-CBR. It's important to recognize, however, that symbolic AA-CBR is a very specific approach. Even though the improvements are significant, they may only have a limited impact, audience wise.

## Related Work

### Builds on Argumentation and Neurosymbolic Learning

A satisfactory summary of the relevant research is given in the paper. The debate though, could be more critical. Instead of just outlining the ways in which Gradual AA-CBR is different from current approaches, the authors could have done a more thorough examination of its advantages and disadvantages.

### Compares to AA-CBR and NAL

Though slightly biased, the comparison to AA-CBR and NAL is important.   They mostly highlight AA-CBR's disadvantages and how it differs from NAL, but they don't completely recognize any potential benefits of these different approaches.   It would have been nice to see a more balanced perspective.

### Draws Inspiration from Graph Structure Learning

Graph Structure Learning (GSL) has an interesting and important connection. How does Gradual AA-CBR measure up against the most advanced GSL methods? Is it possible to transfer any knowledge between these fields?

## Experiments & Results

### Evaluated on Classification Tasks

Although the evaluation on standard classification datasets is required, it is also somewhat baseline. Although the dataset selection is practical, it doesn't push the model to its maximum limit.   Seeing the model tested on challenging or real-world datasets would have been a stronger argument.

### Compared to Neural Networks and AA-CBR Variants

Although the comparisons to neural networks and AA-CBR variations are important, caution should be used when interpreting the results. Although Gradual AA-CBR has "comparable performance" to NNs, this does not imply that it is better in application.

### Demonstrates Comparable Performance to NNs

More investigation is required regarding the "comparable performance" claim. Is it comparable based just on accuracy? What about robustness, training time, or computational cost? A broader understanding would result from a more thorough examination of all of these factors.

## Pros and Cons

### Pros:

* Quality: Although there are few places where the method and experiments could have been more thorough, overall the paper quality is good.

* Clarity: Though the formal terminology can be complex and challenging to understand, the paper is generally well-written.

* Originality: Both the design and the learning process of the framework are innovative.

Significance:
* Although the work could advance explainable AI, resolving the limitations is necessary to ensure its real-world applicability.
* Combines neural networks and argumentation to enhance accessibility.
gains knowledge of how a case-based debate is organized.
* Manages continuous data and categorisation into several classes.
* Understands the significance of features and data points automatically.

### Cons:

* The model's sensitivity to initialization is a significant flaw that may restrict its reliability and ease of use.
* Despite being common in neural networks, the risk of local minima requires a greater discussion and preventative measures.
* Scalability issues can prevent the framework on being used for large scale problem solutions.

**Anonymity:**

Remain anonymous

---

### Official Review · Reviewer_MwBy · 2025-04-06

**Rating:** 7
**Confidence:** 3

**Review:**

This submission proposes Gradual Abstract Argumentation for Case-Based Reasoning (Gradual AA-CBR), in which the classification is determined by an explicit argumentation network, whose weights are learned with neural extractors. In this method, every data point in the training set argues in favor of its labeling. The paper brings three contributions: the Gradual AA-CBR method, its design and implementation as an end-to-end learnable framework, and its evaluation on four datasets.

The paper is well-written and accessible. I found its storyline convincing, though computational argumentation is not my area of expertise. The contributions are well-supported by the exposition and the evaluation of the method. Overall, I see the value of this work being presented and discussed at NeSy.

My main request for clarification concerns the evaluation. The authors claim that the results are positive because there are "only small margins of performance lost due to the added interpretability and transparent reasoning mechanism". I have two concerns with this claim. First, the connection between model accuracy and interoperability as a tradeoff is disputable (and disputed, see the popular survey in [1] for one discussion). Thus, claiming that the performance loss can be justified by added interpretability and transparency needs support. From the paper, this tradeoff is not clear at present.
Second, while CBR is an interpretable method by definition, and the argumentative network is indeed transparent, it is unclear whether this method's interpretability and transparency are meaningful. It would help if the authors could include a quantitative evaluation, or at least a qualitative study, that shows what kind of interpretability and transparent reasoning is gained in practice over NNs.

My other concern regarding the evaluation is that it evaluates Gradual AA-CBR as a whole against other systems. I understand that space limitations prohibit extensive ablations, but to support the method's effectiveness claim, it would be helpful to understand how it performs, e.g., when some, but not all three, of the edge weighting mechanisms are used.

Finally, given that the NN's task performance is already very high (nearly 100%) on these tasks, it raises the question of how the proposed method would perform on more complex classification tasks. A brief discussion about what it would take for the method to generalize to more complex tasks would enrich the paper.

**Anonymity:**

Remain anonymous